# Seedling recruitment under isolated trees in a tea plantation provides a template for forest restoration in eastern Africa

Henry J. Ndangalasi[1], Cristina Martínez-Garza[2], Tesakiah C. A. Harjo[3], Clayton A. Pedigo[3], Rebecca J. Wilson[3], Norbert J. Cordeiro[3,4]*

**1** Botany Department, University of Dar es Salaam, Dar es Salaam, Tanzania, **2** Departamento de Ecología Evolutiva, Centro de Investigación en Biodiversidad y Conservación, Universidad Autónoma del Estado de Morelos, Cuernavaca, Morelos, México, **3** Department of Biology, Roosevelt University, Chicago, Illinois, United States of America, **4** Negaunee Integrative Research, Science & Education, The Field Museum, Chicago, Illinois, United States of America

\* ncordeiro@fieldmuseum.org

**Data Availability Statement:** Data has been uploaded to KNB and is publicly available here: https://knb.ecoinformatics.org/view/doi%3A10.5063%2FFX77T3.

## Abstract

Natural regeneration is less expensive than tree planting, but determining what species will arrive and establish to serve as templates for tropical forest restoration remains poorly investigated in eastern Africa. This study summarises seedling recruitment under 29 isolated legacy trees (14 trees comprised of three exotic species and 15 trees comprised of seven native species) in tea plantations in the East Usambara Mountains, Tanzania. Among the findings were that pioneer recruits were very abundant whereas non-pioneers were disproportionately fewer. Importantly, 98% of all recruits were animal-dispersed. The size of legacy trees, driven mostly by the exotic *Grevillea robusta*, and to some extent, the native *Milicia excelsa*, explained abundance of recruits. The distribution of bird-dispersed recruits suggested that some bird species use all types of legacy trees equally in this fragmented landscape. In contrast, the distribution of bat-dispersed recruits provided strong evidence that seedling composition differed under native versus exotic legacy trees likely due to fruit bats showing more preference for native legacy trees. Native, as compared to exotic legacy trees, had almost two times more non-pioneer recruits, with *Ficus* and *Milicia excelsa* driving this trend. Implications of our findings regarding restoration in the tropics are numerous for the movement of native animal-dispersed tree species in fragmented and disturbed tropical forests surrounded by farmland. Isolated native trees that bear fleshy fruits can attract more frugivores, resulting not only in high recruitment under them, but depending on the dispersal mode of the legacy trees, also different suites of recruited species. When selecting tree species for plantings, to maximize visitation by different dispersal agents and to enhance seedling recruit diversity, bat-dispersed *Milicia excelsa* and *Ficus* species are recommended.

**Funding:** Material support was provided by University of Dar es Salaam (Botany Department), Roosevelt University and Universidad Autónoma del Estado de Morelos. HJN and NJC were supported by the Carnegie African Diaspora Fellowship Program. The funders had no role in study design, data collection and analysis, decision to publish, or preparation of the manuscript. No authors received salary from any of the funders. The authors received no specific funding for this work.

**Competing interests:** The authors have declared that no competing interests exist.

# Introduction

Tropical forests have been modified by human disturbance for centuries, but in the Anthropocene era, habitat loss and fragmentation have escalated to the point that once extensive forests are now dissected landscapes. In the Eastern Afromontane biodiversity hotspot of eastern Africa, habitat fragmentation and loss are considered among the two greatest threats to biodiversity [1]. For example, the East Usambara Mountains, Tanzania, are one of the most species and endemic rich sites in this biodiversity hotspot, but extensive tracts of forest were deforested and converted to tea plantations during the British colonial administration when tea was introduced from India as a cash crop [2]. The post-colonial period has added further threats to the forest in the East Usambaras, including subsistence agriculture which has encroached into the protected and unprotected forest [2]. Deforestation and forest fragmentation has resulted in declining abundances, and some cases, local extinctions of species, such as for birds [2]. It has also contributed to diminished mutualisms between animal seed dispersers and their dependent plants [3]. Establishing planted restoration corridors linking key forest fragments with the continuous forest [4], a conservation strategy with strong evidence of success [5], has been advocated to mitigate or forestall this biodiversity collapse. However, in eastern Africa, experimental forest restoration is scant to non-existent, and knowledge is limited on the restoration process and potential tree species to plant.

Ecological restoration seeks to recover the attributes, function, and conditions of damaged ecosystems [6] but in permanent agricultural landscapes, full recovery is not possible. In such cases, forest landscape restoration is suggested to create an equilibrium of natural and productive areas [7]. To successfully develop restoration treatments and prioritize areas to restore worldwide, the need to measure the potential for natural regeneration is recommended for the entire landscape [8]. In this context, tea plantations may remain productive, and some areas of the plantation can be used for conservation of tree species or be converted to planted restoration corridors that would favour the movement of animals and the plants they transport across the landscape [9]. Minimal restoration intervention [10, 11] or unassisted forest regeneration [12] is suggested when a high potential for natural regeneration is identified, which reduces restoration costs [13]. Therefore, potential for natural regeneration in tea plantations surrounded by natural forest could be evaluated to further identify areas for restoration actions or to develop a template of native tree species likely to recruit naturally in restoration corridors. Developing such a recruitment template requires an understanding of interactions among the tree species that can be dispersed, their dispersal vectors, and the landscape characteristics.

Successful arrival and establishment of trees in restored sites depends on barriers associated with dispersal and establishment limitations. Seed dispersal is a key phase in plant regeneration processes and serves as a recruitment template [14]. Dispersal limitation refers to the failure of a seed to reach a site, while establishment limitation reflects the failure of arriving seeds to germinate, survive and grow [15, 16]. Seeds are often transported at different distances away from parent trees, depending on the dispersal vector: for example, wind-dispersed seeds may sometimes move long distances in open areas [17]. Seeds dispersed by animals will move different distances depending on the animal taxon moving them. For example, small birds generally distribute seeds short distances, whereas mammals and medium-sized birds disperse seeds at longer distances [18]. Also, in agricultural landscapes, higher levels of dispersal limitation have been found for animal than for wind-dispersed rainforest tree species [16, 19]. More generally, seed dispersal is expected to decrease with distance from vegetation border [19, 20], but given that many animals may not move frequently outside the forest [21–23], more seeds dispersed by wind are expected to arrive into open habitats.

Life history attributes of tree species contribute to seed dispersal and recruitment in degraded areas. Pioneer tree species, usually characterized by small seeds, tend to move easily by wind or by the many animals they attract [24, 25]. These pioneers are fast-growing species and are often the

first to colonize open fields adjacent to forest [26, 27], and the edges of continuous forest or small forest fragments [28]. On the other hand, non-pioneer tree species generally have larger seeds which are usually dispersed by few specialist animals [29], and are transported at comparatively shorter distances than their pioneer counterparts [30]. In degraded areas, non-pioneer species have higher dispersal limitation than pioneer species, and all but a few pioneer species show high establishment limitation [16]. Seedling establishment in abandoned agricultural areas is also limited by resource competition and microclimate conditions [31]. Improved environmental conditions under isolated trees or shrubs in such fields may increase tree recruitment [32, 33], but given higher dispersal limitation for non-pioneer as compared to pioneer species, more of the latter are expected to successfully arrive and establish under isolated trees outside the forest.

In most tropical agricultural landscapes adjacent to forest, some native forest trees are retained, or sometimes, exotic trees are planted to provide shade for various crops. This is true of the tea plantations in eastern Africa as well as those in south-east Asia, where native remnant trees and exotic trees planted during the colonial period, and in some cases thereafter, serve the purpose of providing shade to the tea [34, 35]. These isolated trees often function as recruitment foci for the recovery and regeneration of native forest tree species. For example, in farmland adjacent to forest in western Kenya, fleshy-fruited exotic guava *Psidium guavaja* (Myrtaceae) trees attracted frugivorous birds, leading to higher seed rain and seedling establishment under them [36]. In Mexico, isolated trees in human-made pastures enhanced the arrival and establishment of native tree species [37, 38]. A topic of much deliberation in recent years is the use of exotic tree species to favour forest regeneration [11, 39–41]. In some environments, it is possible that trees with fleshy fruits may act as recruitment foci to attract forest seed-dispersing animals to degraded areas [42], and thus increase the seed rain of native forest trees.

Here, we evaluate richness and abundance of recruits under native and exotic legacy trees in tea plantations in East Usambara Mts, Tanzania. We follow Jacob et al. [11] and Chetan et al. [41] in our use of the term legacy trees to refer to isolated trees which are remnants from the past. We explore which attributes of these legacy trees explain successional status and dispersal mode of the seedling recruitment community underneath them. We expect to find higher richness and abundance of pioneer trees than non-pioneer trees dispersed by animals under legacy trees closer to natural forest given dispersal and establishment limitations. Recent reviews indicate that the potential for natural regeneration is higher closer to native vegetation remnants, on steep slopes, adjacent to protected areas, and far from heavily populated areas [43, 44]. Furthermore, flying animals may be attracted to legacy trees due to their fleshy fruits or because of tree height and canopy cover [35, 45]. They may also use these trees as stepping stones to rest as they move through these dissected landscapes. As such flying animals use these legacy trees, they have the potential to disperse many seeds, and in some cases of multiple species, increasing the establishment of forest tree species under them. We expect different recruitment composition among legacy trees based on tree size, availability of fleshy fruits, distance to the nearest forest, and whether they are native or exotic trees. Measuring richness and abundance of recruits under native and exotic legacy trees will help pinpoint key species that are dispersed and therefore have high recruitment potential. Furthermore, by considering aspects of the landscape context for seedling recruitment of native tree species, this study can provide a baseline framework from which to plan forest restoration in this region.

## Methods

### Study area

The East Usambara Mountains are part of the globally biodiverse Eastern Arc Mountains of Tanzania and Kenya. Forest cover in the southern part of the East Usambaras starts at about

250 m and is continuous to 1100 m above sea level. Annual rainfall averages 2000 mm per year. While precipitation is largely seasonal, falling from late March to May and again from October to November, additional orographic rainfall occurs in other months due to the proximity to the Indian Ocean [46]. The exception is January and February, which are the hottest months. The average temperature is approximately 24°C.

At elevations of about 900 to 1100 m on the plateau, there are two large tea plantations that were established during the colonial period [2], leading to much of the primary submontane forest to have been destroyed and fragmented. Remaining forest in the south to central part of the East Usambaras is now protected as the 8360 ha Amani Nature Reserve. Numerous forest fragments of varying sizes occur within the tea plantations, most originating from when the plantations were first established. Consistent presence of cloud forest at higher elevations maintains a wet forest type, with several dominant tree species occurring throughout the submontane forest of Amani Nature Reserve and surrounding forest fragments. These include at least four emergent species: *Newtonia buchananii* (Fabaceae), *Maranthes goetzeniana* (Chrysobalanaceae), *Pouteria adolf-friedricii* (Sapotaceae), and *Parinari excelsa* (Chrysobalanaceae); numerous canopy and midstorey trees such as *Cephalosphaera usambarensis* (Myristicaceae), *Isoberlinia schefflerii* (Fabaceae), *Macaranga capensis* (Euphorbiaceae), *Greenwayodendron suaveolens* (Annonaceae), *Leptonychia usambarensis* (Sterculiaceae), *Myrianthus holstii* (Cecropiaceae), *Allanblackia stuhlmannii* (Clusiaceae), *Sorindeia madagascariensis* (Anacardiaceae), *Synsepalum cerasiferum* (Sapotaceae), *Anisophyllea obtusifolia* (Anisophylleaceae), and *Strombosia scheffleri* (Olacaceae); and an exotic, invasive gap- and edge-specialist, *Maesopsis eminii* (Rhamnaceae), which is also a dominant canopy species in the forest [46].

## Sampling

Within the tea plantations are various native legacy tree species (Figs 1 and 2), and at least two exotic tree species: *Grevillea robusta* and *Albizia* sp. Both of these exotics were planted in East African tea plantations as shade trees [34]. Over the last 20 years, two of us (NJC and HJN) observed native tree species recruiting under these legacy trees. The presence of large-seeded species on occasion, as well as observations of the high diversity of seedling recruits, provided the impetus to quantitatively evaluate the role that legacy trees could have towards future restoration efforts.

In May 2014, just after the heavy rainy season, we targeted isolated, legacy trees that occurred in the tea plantations north of Amani Nature Reserve, in the area of Monga (Figs 1 and 2). We selected 15 trees, six exotics and nine natives (Table 1). In May 2018, we selected 14 new trees in the Mbomole-Derema area (Figs 1 and 2), and these included eight exotics and six natives (Table 1). For each focal tree, we measured the distance from the trunk to the crown edge at each of the cardinal positions to obtain the crown area (Table 1). Other measurements taken were the diameter at breast height (DBH) for the girths, and the heights of the legacy trees. Heights were measured with a Bushnell Sport 850 Rangefinder. We used the same rangefinder to measure the distance of the focal legacy tree to the nearest reproductive conspecific, and when none was found within the vicinity, we used the distance to the nearest forest fragment as a conservative estimate (see S1 Table). We searched the forest fragment for conspecifics of the species of recruits to be certain of their presence there.

To sample the seedling and sapling recruits, we marked a circle of a fixed radius of 5 m around the trunk of each legacy tree. We divided this circle into wedges to make it easier to sample all recruits. This was also necessary because we often had to move away the tea bushes to locate seedlings and sapling. Recruits were classified as seedlings if they had leaves that had freshly emerged from drying up cotyledons, whereas saplings were < 1 m in height. Larger

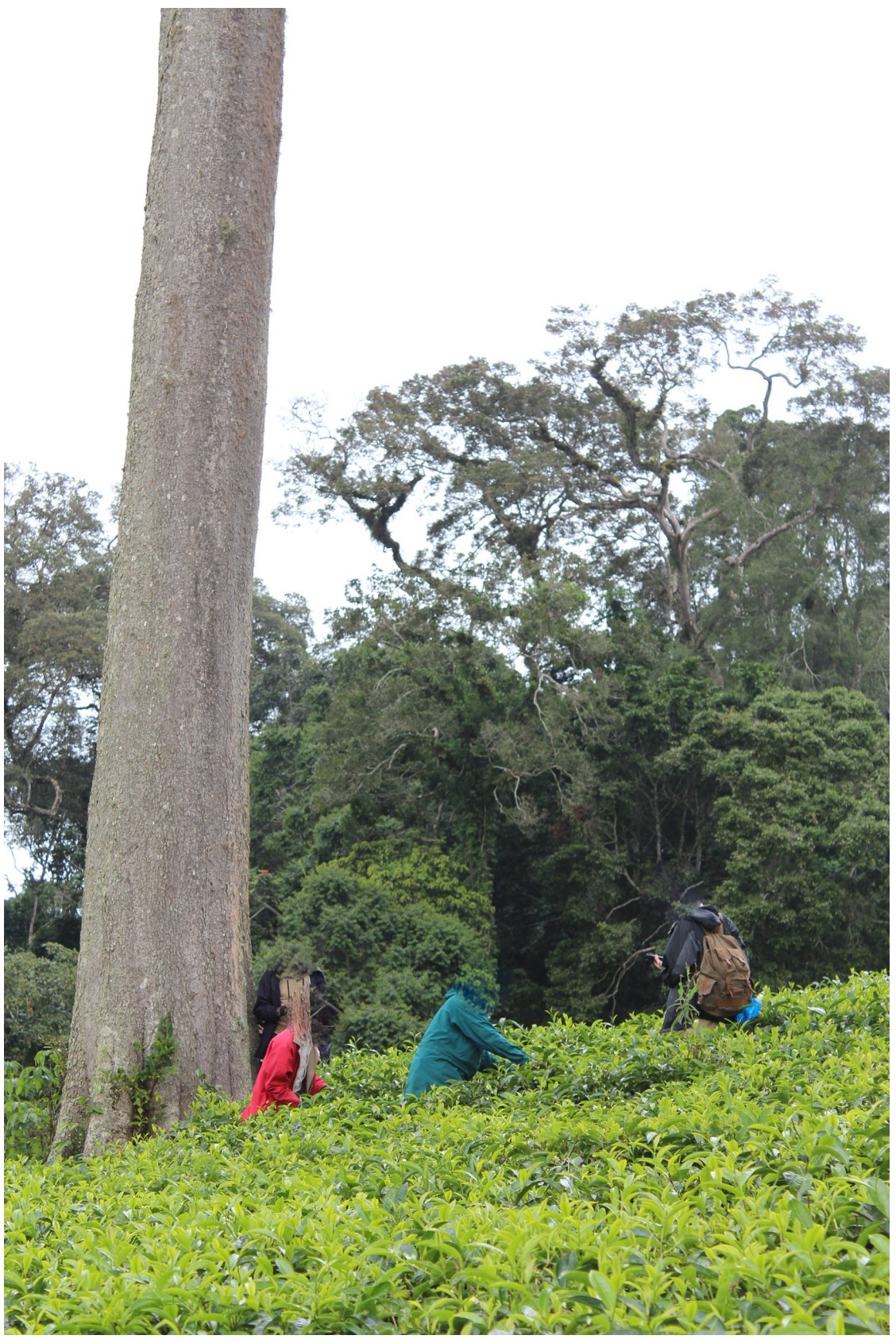

**Fig 1. Photograph of one of the large *Parinari excelsa* (Chrysobalanaceae) trees sampled in the tea plantation near Monga in the East Usambara Mts, Tanzania.**

saplings are often removed by the tea workers on an annual basis and we can therefore assume that recruits are generally not older than one-year old. We identified and counted all the recruits under the legacy trees. Permission to conduct this study was obtained from the Tanzania Commission for Science and Technology (COSTECH), Amani Nature Reserve, and the East Usambara Tea Company.

## Data analysis

Fourteen recruits that were not identified to species were excluded from analyses. Conspecific seedling recruits under legacy trees, which would have most likely arisen from fallen, un-dispersed seeds from parent trees, were also excluded from the analyses. A Principal Component Analysis (PCA) was run to ordinate six variables of 29 isolated trees (Table 2). The use of PCA to integrate attributes of legacy trees into two axes allows to reduce the number of correlations, as these axes can then be used to predict richness and abundance of recruits under isolated trees with regressions. Additionally, recruits were categorized by successional status (pioneer vs non-pioneers) and one analysis included recruits of species dispersed by animals. Non-

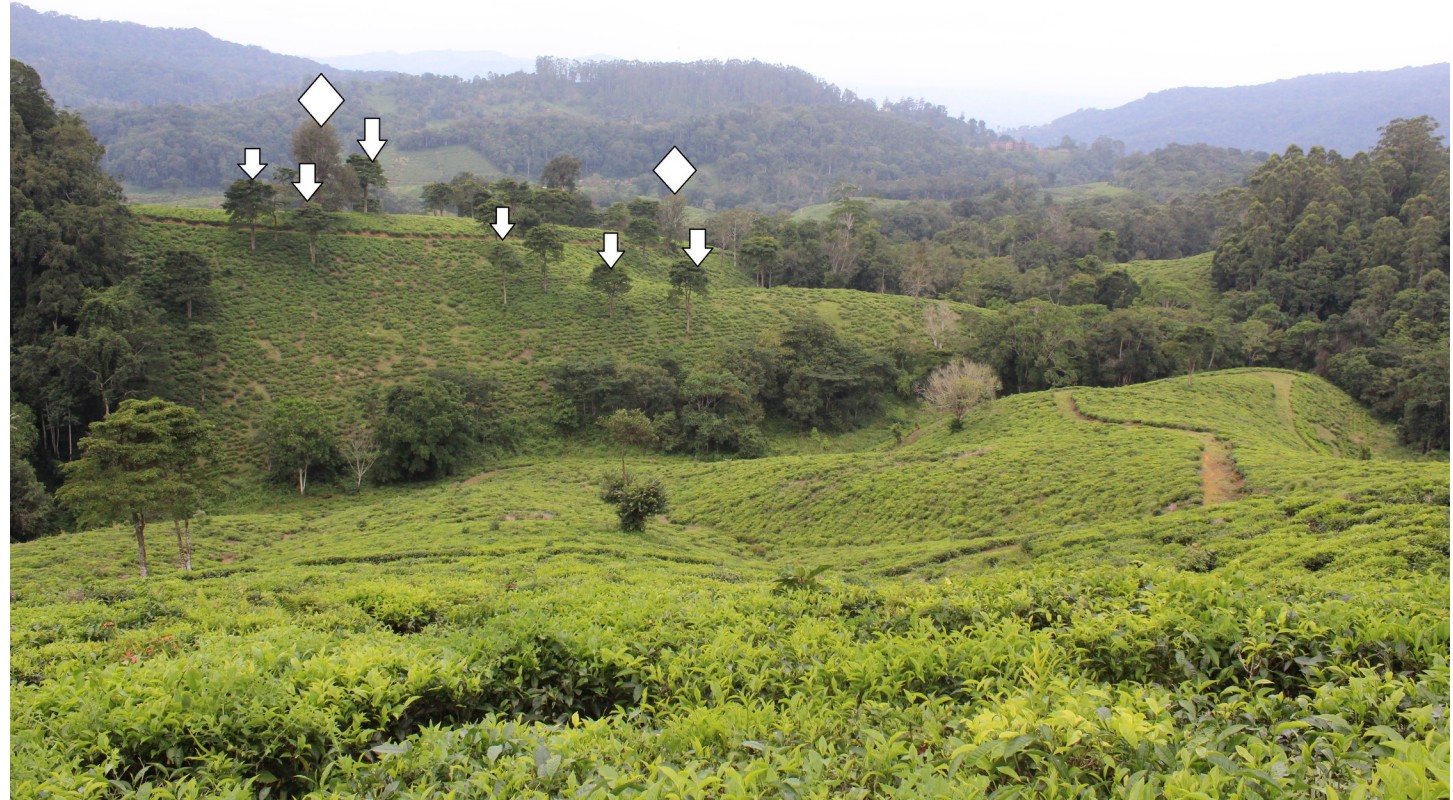

**Fig 2. Photograph showing a cluster of *Milicia excelsa* (arrows) and *Grevillea robusta* (diamonds) legacy trees in the tea plantation near Mbomole-Derema in the East Usambara Mts, Tanzania.**

**Table 1. General characteristics of the 29 legacy trees, including seed dispersal mode, year sampled, height, DBH and crown area in tea plantations in the East Usambara Mts, Tanzania.**

| Legacy species | Height (m) | DBH (cm) | Crown area (m²) | Dispersal mode | Code name |
|---|---|---|---|---|---|
| **Exotic** | | | | | |
| *Albizia* sp | 16 | 101 | 599.37 | abiotic α | Alsp |
| *Albizia odoratissima* φ | 13 | 66.8 | 130.7 | abiotic α | Alod |
| *Grevillea robusta* | 7 | 71 | 80.12 | abiotic α | Grro |
| *Grevillea robusta* | 27 | 71.6 | 136.85 | abiotic α | Grro |
| *Grevillea robusta* | 22 | 116.3 | 167.99 | abiotic α | Grro |
| *Grevillea robusta* | 29 | 118.4 | 186.87 | abiotic α | Grro |
| *Grevillea robusta* | 22 | 104.6 | 221.01 | abiotic α | Grro |
| *Grevillea robusta* φ | 18 | 71 | 121.74 | abiotic α | Grro |
| *Grevillea robusta* φ | 16 | 58.3 | 58.56 | abiotic α | Grro |
| *Grevillea robusta* φ | 16 | 95.5 | 155.04 | abiotic α | Grro |
| *Grevillea robusta* φ | 17 | 71.9 | 194.83 | abiotic α | Grro |
| *Grevillea robusta* φ | 22 | 114.6 | 206.12 | abiotic α | Grro |
| *Grevillea robusta* φ | 15 | 74.2 | 103.24 | abiotic α | Grro |
| *Grevillea robusta* φ | 17 | 55.7 | 93.31 | abiotic α | Grro |
| **Native** | | | | | |
| *Anisophylea obtusifolia* | 18 | 86.1 | 112.16 | animal *†∞† | Anob |
| *Anthocleista grandflora* | 13 | 116 | 175.54 | animal † | Angr |
| *Ficus thonningi* | 18 | 118.9 | 225.65 | animal*†∞‡ | Fith |
| *Ficus* sansibarica | 12 | 133.1 | 200.43 | animal*†∞‡ | Fisa |
| *Parinari excelsa* | 32 | 115 | 129.99 | animal *†∞† | Paex |
| *Parinari excelsa* | 26 | 143.3 | 632.36 | animal *†∞† | Paex |
| *Parinari excelsa* | 31 | 128.8 | 332.48 | animal *†∞† | Paex |
| *Parinari excelsa* | 21 | 115.4 | 285.02 | animal *†∞† | Paex |
| *Pouteria adolf-friedricii* | 23 | 93.3 | 27.11 | animal †∞‡ | Poad |
| *Milicia excelsa* φ | 20 | 118 | 284.28 | animal † | Miex |
| *Milicia excelsa* φ | 21 | 134.3 | 194.83 | animal † | Miex |
| *Milicia excelsa* φ | 15 | 71.6 | 82.52 | animal † | Miex |
| *Milicia excelsa* φ | 17 | 78 | 151.09 | animal † | Miex |
| *Milicia excelsa* φ | 12 | 55.5 | 98.96 | animal † | Miex |
| *Milicia excelsa* φ | 24 | 137 | 326.85 | animal † | Miex |

Year sampled: no symbol = 2014; φ = 2018

Abiotic: α = ballistic and/or wind

Animal dispersal

* = bird

† = bat

∞ = primate

‡ = African palm civet *Nandinia binotata*

† = African giant rat *Cricetomys gambianus*

pioneer species are basically shade-bearers or late-successional species. The PCA was performed in STATISTICA 7.0 [47].

To analyse composition of recruits under native and exotic trees, we performed non-metric multidimensional scaling (NMDS) analyses [48]. The species similarity metric used was Pearson correlations with recruit abundance under each legacy tree, and the stress value in the NMDS was used to determine the ordination fitness [49]. The number of NMDS dimensions

**Table 2. Correlation of PCA axis 1 and 2 and richness and abundance of seedling recruits under 29 isolated trees in tea plantations in the East Usambara Mountains, Tanzania.**

| Variable/ axis | PCA Axis 1 | PCA Axis 2 |
|---|---|---|
| Richness | -0.2 | -0.3 |
| Abundance | 0.35 | -0.42* |
| Richness–dispersed by animals | -0.23 | -0.23 |
| Abundance–dispersed by animals | 0.35 | -0.41* |
| Richness–pioneers | -0.02 | -0.1 |
| Abundance–pioneers | 0.37* | -0.39 |
| Richness–non-pioneers | -0.37 | -0.43* |
| Abundance–non-pioneers | -0.23 | -0.41* |

($^*$ = p < 0.05).

was decided based on the lowest stress value (the lower the stress value, the better the fit) and the more meaningful solution; d-hat stress values are reported [47]. We first conducted an NMDS on the abundance of all recruit species under the legacy trees. We then classified the recruit species into six main dispersal modes following [50] and Cordeiro & Ndangalasi (unpublished data): (1) bird-primate-bat, (2) bat, (3) bird, (4) rodent, (5) wind and (6) ballistic dispersal.

## Results

### Species richness, abundance, and guilds of recruits

Of a total of 2800 seedling recruits, there were 38 and 47 species represented under exotic and native legacy trees, respectively (S2 Table). Fourteen recruits (0.5%) were not included in analyses of dispersal mode of recruits because they were unidentified to species level and therefore of unknown dispersal mode or successional status. The majority of seedling recruits were dispersed by animals (2738 seedlings; 98.2%), whereas the remainder were abiotically dispersed (48 seedlings of five species dispersed by wind, gravity or ballistically; 1.7%). When evaluating patterns of recruitment based on successional status, a very high number of pioneers (2784 recruits of 28 species) was observed, and relatively few non-pioneers (202 recruits represented by 22 species). Fourteen recruits were unidentified to species, whereas for two identified species, the successional status was not known. It is noteworthy that an exotic, invasive pioneer tree species, *Maesopsis eminii* (Rhamnaceae), dispersed by animals, accounted for 30.1% of all recruits, with 676 recruits under native legacy trees (45.1 ± 18.6, mean ± SE per tree) and 167 recruits under 12 of 14 exotic legacy trees (13.9 ± 3.5 per tree).

An examination of the mean number of recruited species under exotic and native legacy trees was 9.1 ± 1.1 (SE) and 11.1 ± 1.1, respectively, whereas the mean number of recruits was almost comparable between the two legacy tree types: 98.3 ± 27.2 for exotic and 94.9 ± 19. 6 for native trees (Table 3). Generally, mean richness and abundance were similar for categories related to seed dispersal mode and successional status, including animal-dispersed, pioneer and non-pioneer recruits (Table 3). The most species and recruits were found under *Grevillea robusta*, the two *Ficus* species, and *Milicia excelsa* (Table 3). Native legacy tree species, such as *Milicia excelsa* and *Ficus* species more specifically, accounted for almost two times more non-pioneer species and recruits (Table 3).

Six traits of the 28 isolated, legacy trees were evaluated for the two sampled years (Table 1). In the PCA analysis of six attributes of the legacy trees, the first two axes explained 65.33% of the variation in legacy trees attributes (Fig 3). The PCA axis 1 was correlated to decreasing DBH, whereas axis 2 was correlated to increasing distance from forest and dispersal vector

**Table 3. Mean number (±SE) of species (Richness) and abundance (Abund; number of recruits) of seedling recruits under exotic and native legacy trees in tea plantations in the East Usambara Mts, Tanzania.**

| Legacy Tree | Richness total | Abund total | Richness A-disp | Abund A-disp | Richness pioneer | Abund pioneer | Richness non-pio | Abund non-pio |
|---|---|---|---|---|---|---|---|---|
| **Exotic** | | | | | | | | |
| *Albizia* (2) | 7.5±2.5 | 48.0±8.5 | 6.0±2.1 | 46.5±8.1 | 5.5±2.5 | 44±7.1 | 1.5±0.4 | 3.5±1.8 |
| *Grevillea* (12) | 10.3±1.2 | 106.7±31.1 | 9.5±1.0 | 106.7±32 | 7.5±0.7 | 101.3±30.3 | 2.2±0.5 | 4.7±1.2 |
| **Native** | | | | | | | | |
| *Ficus* (2) | 12±2.8 | 101.5±19.4 | 12.0±2.8 | 101±19.1 | 7.5±1.1 | 86.0±14.8 | 4.5±1.8 | 15.5±4.6 |
| *Parinari* (4) | 10.0±1.6 | 39.3±9.3 | 9.0±1.1 | 37.0±9.6 | 7.0±0.6 | 35.0±9.3 | 2.81.1 | 3.50.8 |
| *Milicia* (6) | 12.8±1.8 | 134.3±37.1 | 11.7±1.9 | 130.3±36.0 | 7.7±1.1 | 122.0±37.0 | 4.7±0.6 | 11.8±3.0 |
| Other sp. (3) | 8.3±1.7 | 86.0±36.2 | 8.0±1.4 | 85.7±35.9 | 5.3±1.2 | 78.3±36.4 | 2.7±0.3 | 7.3±1.9 |
| Exotic (14) | 9.9±1.1 | 98.3±27.2 | 9.0±1.0 | 98.1±28.0 | 7.2±0.7 | 93.1±26.5 | 2.1±0.4 | 4.5±1.0 |
| Native (15) | 11.1±1.1 | 94.9±19.6 | 10.3±1.0 | 92.6±19.1 | 7.0±0.6 | 85.3±19.1 | 3.7±0.5 | 9.2±1.8 |

Notes: Legacy tree species grouped by genus (refer to Table 1 for full species name) and by exotic versus native.

Recruits are grouped into three categories (animal-dispersed [A-disp], pioneer and non-pioneer [non-pio]) as a function of the genus of legacy tree species to identify preliminary trends in recruitment potential. Native legacy trees categorized as "Other sp." include *Anisophyllea obtusifolia*, *Anthocleista grandiflora* and *Pouteria adolf-friedricii*.

(Figs 3 and 4). More specifically, the PCA shows that the exotic tree, *Grevillea robusta* (Proteaceae), occurred at longer distances from the nearest forest fragments or continuous forest compared to the native tree, *Milicia excelsa* (Moraceae) (Fig 4). These two legacy trees clearly separated from the cluster based on their seed dispersal mode and girth (DBH). *Grevillea robusta* is dispersed by wind and *Milicia excelsa* by bats, and whereas both species showed a similar variation in DBH size, *Milicia excelsa* was smaller on average, with all other legacy tree species occurring in between.

PCA axis 1 was positively correlated to the abundance of pioneer recruits (Fig 5, Table 2). PCA axis 2 was negatively correlated to the abundance of all recruits (Fig 6, Table 2), abundance of recruits dispersed by animals (Fig 7) and abundance of non-pioneer recruits (Fig 8; Table 2). The abundance of pioneer recruits appears to be driven by the average large girth of *Grevillea robusta* trees (Fig 5). Therefore, repeating this PCA without *Grevillea robusta* produced a slighter higher $R^2$ but the relationship was not significant ($R^2 = 0.15$, p = 0.11). For the PCA on the abundance of all recruits (Fig 6) and only animal-dispersed recruits (Fig 7) under legacy trees, the farther away from the nearest forest fragment or the edge of continuous forest leads to fewer recruits. This is supported by ordination of attributes of legacy trees (Fig 5) in which *Grevillea robusta* is farthest from the nearest forest. Furthermore, *Milicia excelsa*, the native legacy tree species dispersed by bats (Fig 5), has the highest abundances of all recruits on average (Fig 6) as well as animal-dispersed recruits (Fig 7). The correlation for the abundance of non-pioneer recruits matches those for all recruits and pioneer recruits, with distance from nearest forest source and dispersal vector explaining the highest variation (Table 2). However, abundance of non-pioneer recruits is expectedly very low, being about 10 times less than for pioneer recruits (Fig 8).

## Community composition of seedling recruits

The Non-metric multidimensional scaling (NMDS) ordination of the recruited species classified by dispersal mode was meaningful with two dimensions (stress = 0.14). Recruit composition was highly homogeneous under the six bat-dispersed *Milicia excelsa* legacy trees. Recruit composition under the 11 legacy trees of the exotic *Grevillea robusta* was highly heterogeneous,

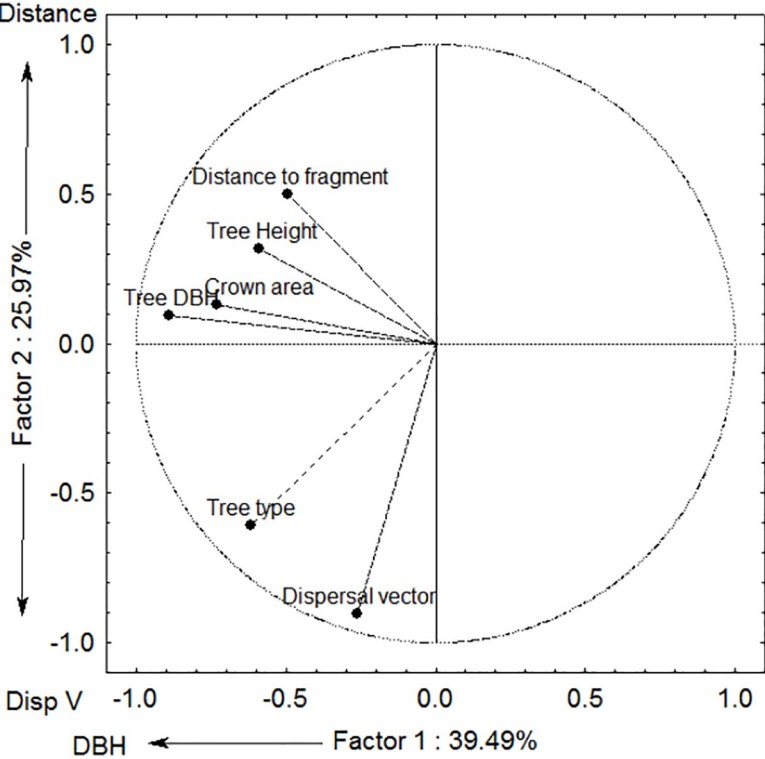

**Fig 3. Trait loading of PCA axes 1 and 2 of an ordination based on six attributes for 29 legacy trees in tea plantations in East Usambara Mountains, Tanzania.**

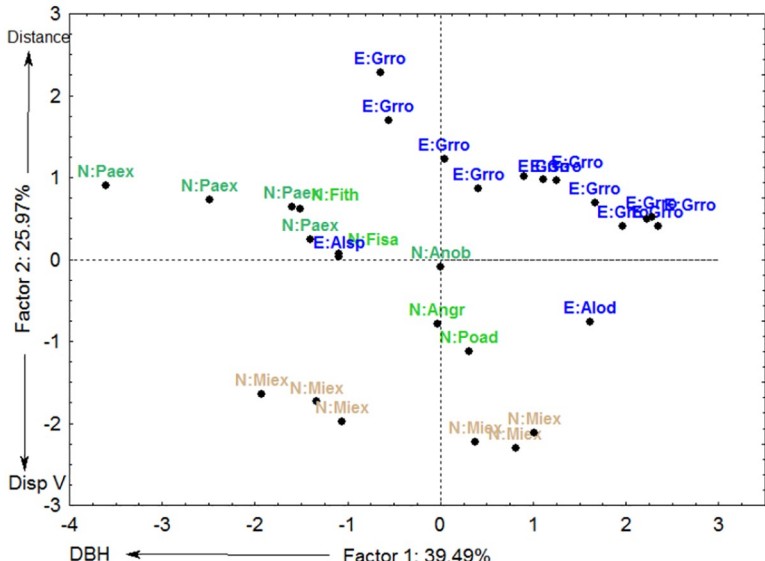

**Fig 4. Plot scores of the PCA axes 1 and 2 of an ordination based on six traits for 29 legacy trees in tea plantations in East Usambara Mountains, Tanzania.** Exotic trees are shown in blue font and natives in two types of green font, the khaki green emphasizing *Milicia excelsa*. Abbreviations for tree species names are in Table 1.

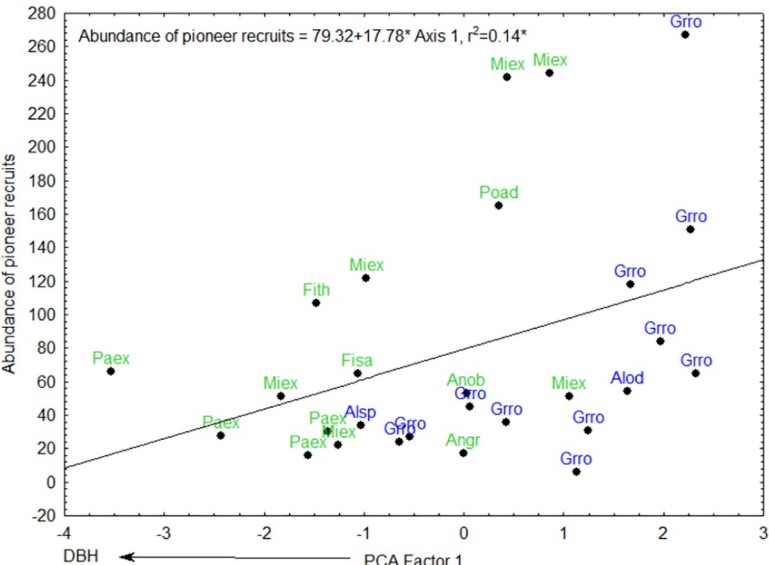

**Fig 5. Regression of abundance of pioneer recruits under 29 native (green) or exotic (blue) legacy trees as a function of PCA axis 1.** Value of $R^2$, regression line, and equation are shown. Abbreviations for tree species names are in Table 1.

overlapping with recruit composition under other legacy trees (Fig 9). For those recruit species dispersed by birds only, the NMDS showed overlapping compositions of species (Fig 10; stress = 0.11) under all legacy trees. For those recruit species dispersed by birds, primates, or bats, four primary groupings emerged (Fig 11). The exotic legacy tree, *Grevillea robusta* in the upper left, separated from the native *Milicia excelsa* in the upper right, and *Parinari excelsa* in the lower left, whereas all other native legacy tree species clustered together in the middle (Fig 11; stress = 0.13). Noteworthy here is that seven legacy trees did not have recruits dispersed by

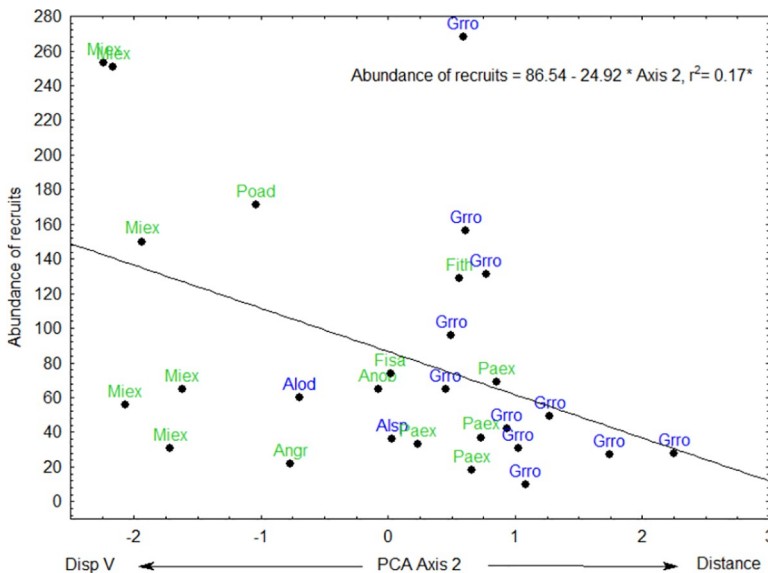

**Fig 6. Regression of abundance of recruits under 29 native (green) or exotic (blue) legacy trees as a function of PCA axis 2.** Value of $R^2$, regression line, and equation are shown. Abbreviations for tree species names are in Table 1.

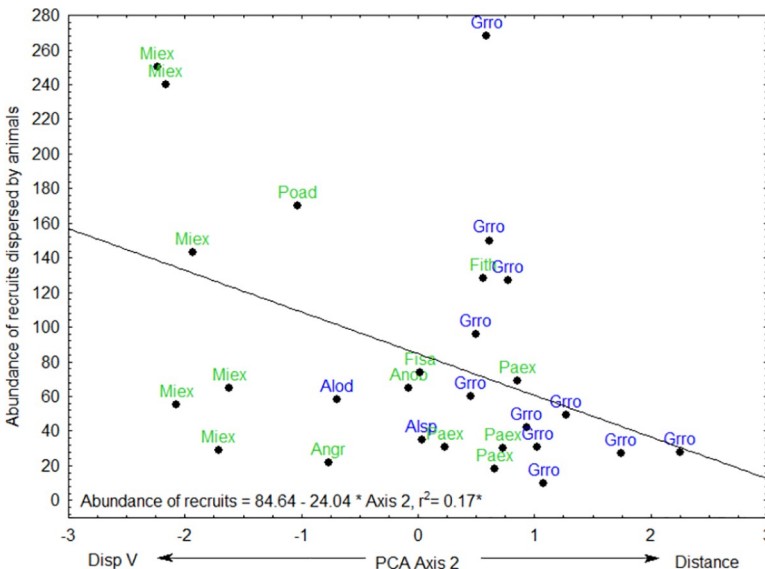

**Fig 7. Regression of abundance of recruits dispersed by animals under 29 native (green) or exotic (blue) legacy trees as a function of PCA axis 2.** Value of $R^2$, regression line, and equation are shown. Abbreviations for tree species names are in Table 1.

birds, primates, or bats (i.e. four *Grevillea robusta*, and one each of *Albizia* sp., *Parinari excelsa* and *Milicia excelsa*).

## Discussion

Isolated trees in monocultures, mixed agroforestry, or abandoned pastures adjacent to forest in the tropics act as recruitment foci for native forest trees. Our study on the recruitment of

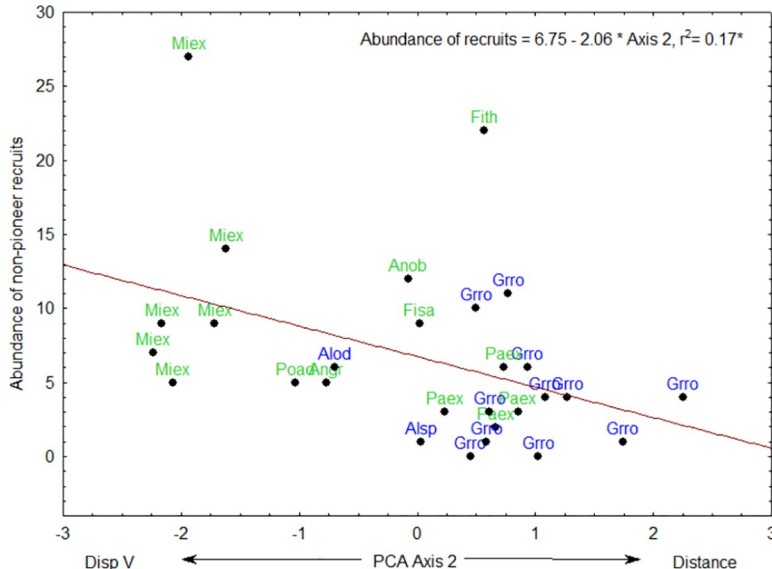

**Fig 8. Regression of abundance of non-pioneer recruits dispersed by animals under 29 native (green) or exotic (blue) legacy trees as a function of PCA axis 2.** Value of $R^2$, regression line, and equation are shown. Abbreviations for tree species names are in Table 1.

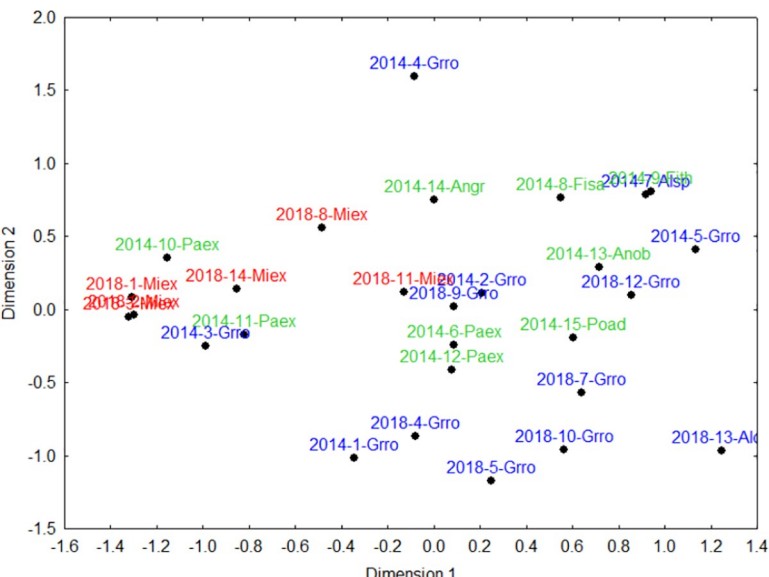

**Fig 9. Non-metric multidimensional scaling (NMDS) analysis for abundance of recruits under 29 native (green), native *Milicia excelsa* (red) and exotic (blue) legacy trees in tea plantations in the East Usambara Mts, Tanzania.** Abbreviations for tree species are in Table 1.

tree seedlings under 29 legacy trees in a tea plantation confirmed results from other studies that isolated trees act as recruitment foci [e.g. 35, 51–54]. Importantly, we found that quite a high number of up to 54 or, at a minimum 50 species (excluding four unidentified), were represented among the 2800 recruited seedlings. The tea plantation surrounds forest fragments and borders sections of continuous forest and the guilds of forest species that recruit under the legacy trees in these plantations provide a useful template to understand the implications for

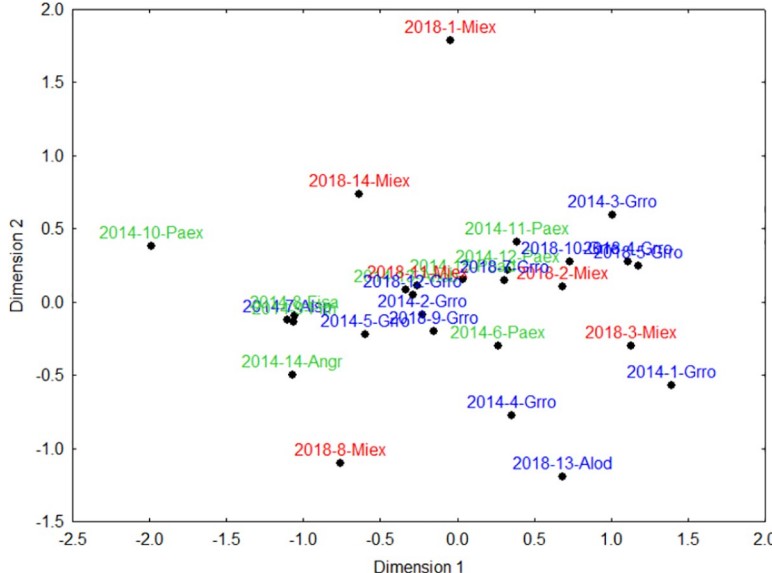

**Fig 10. Non-metric multidimensional scaling (NMDS) analysis for abundance of recruit species dispersed by bird under the native (green, multiple species), native *Milicia excelsa* (red) and exotic (blue) legacy trees in tea plantations in the East Usambara Mts, Tanzania.** Abbreviations for tree species names are in Table 1.

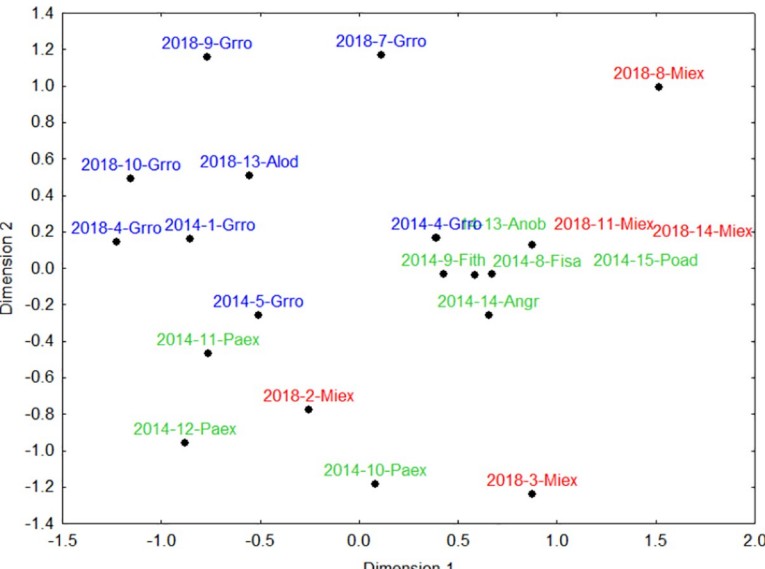

**Fig 11.** Non-metric multidimensional scaling (NMDS) analysis for abundance of recruit species dispersed by birds-primates-bats under the native (green, multiple species), native *Milicia excelsa* (red) and exotic (blue) legacy trees in tea plantations in the East Usambara Mts, Tanzania. Abbreviations for tree species names are in Table 1.

future forest restoration in this region. Given the limited research on tropical forest restoration in eastern Africa, as compared to Central and South America [16, 20, 27, 31, 37, 55, 56], the results from our study add to the handful from Uganda [11, 49, 57, 58] and western Kenya [36] that help to identify native tree species successional pathways or potential pathways towards future restoration efforts.

## Patterns of seedling recruitment

Seedling recruitment occurs when seeds of trees can overcome barriers to seed dispersal. For wind-dispersed species, dispersal limitation is influenced by wind action [17]. On the other hand, for animal-dispersed species, seed or fruit size, fruit type, and fruit crop size are among the factors that affect the fortunes of seed arrival into open areas [e.g. 42, 65]. Distance from source sites to recruitment foci, such as isolated legacy trees, may also play a major role in the types and the numbers of seeds that initiate the recruitment template. In our study, legacy trees like the exotic *Grevillea robusta* and the native *Milicia excelsa* that were generally farther from and nearer to forest on average, respectively, tended to have the highest recruitment. Larger girth size was a strong predictor of increased seedling recruitment, as was found for larger *Ficus* trees that showed higher recruitment potential in a modified landscape in Assam, India [54]. Certainly, increased recruitment under large trees was dominated by the exotic *Grevillea robusta*, but the native bat-dispersed legacy tree species, *Milicia excelsa*, was equally important. In the case of *Milicia excelsa*, this could be explained by being closer to forest on average as well as by attracting frugivorous bats more so than the wind-dispersed *Grevillea robusta* (see below).

Distance from the nearest forest source is a common limitation for seed dispersal and recruitment in more open, agricultural, or successional landscapes [30, 41, 42, 59]. It was therefore of interest that abiotically-dispersed and non-pioneer species were poorly represented under legacy trees. Working in the Philippines, Ingle [19] found that wind-dispersed seeds occurred at 15 times higher densities in successional habitats adjacent to forest than seeds dispersed by bats and birds. While we did not sample seeds, the recruits of wind- and

other abiotically-dispersed forest species were <2% of all recruits, results that were different from Ingle [19] and yet somewhat comparable to studies elsewhere in the tropics [e.g. 59]. In Puerto Rican pastures, seeds of two wind-dispersed species were not found in seed traps beyond 8 m from the forest edge [59], suggesting strong dispersal limitation. Also, in human-made pastures in Mexico, dispersal limitation (failure of seeds to reach a site) ranged from 56 to 99% for five wind dispersed species arriving into restoration settings > 90 m from the closest secondary forest [16]. In the East Usambaras, wind-dispersed *Newtonia buchananii* are common emergent trees whose 5–11 cm long, papery diaspores have been observed being blown >50 m beyond the forest edge (HJN, NJC, pers. obs.). Therefore, to only find single seedlings under three legacy trees could be better explained by high seed establishment limitation following arrival under these isolated trees. For *Newtonia buchananii*, dispersal and establishment limitation needs to be evaluated to better understand the factors that limit the distribution and abundance of recruits under legacy trees. In contrast, another wind-dispersed species, the exotic *Cedrela odorata* (Meliaceae), was found in small numbers under seven legacy trees. Most of the legacy trees were *Milicia excelsa*, which were closest to a small number of *Cedrela* trees that were the likely source of these recruits. Based on evidence of spacing of these *Cedrela* trees, it seems likely that these were planted at the forest edge. However, successful dispersal by wind occurs more frequently in open areas [17] and therefore, in these tea plantations, wind dispersed species may (i) be limited in movement compared to open pastures or early successional environments, or (ii) arrive in numbers, but as has been noted in the Philippines [19] and Mexico [10], may not recruit successfully.

Dispersal by animals and successional status strongly predicted seedling recruitment under legacy trees in East Usambara tea plantations. Pioneer, animal-dispersed species formed the majority of recruits (2738 of 2784 recruits, 28 species), whereas non-pioneer species (202 recruits, 22 species) were poorly represented under legacy trees (Table 3; S2 Table). That the seed rain or seedling recruitment under isolated trees is dominated by mostly animal-dispersed pioneer tree species in successional and agricultural matrix habitats near forest in the tropics is commonly reported [e.g. 20, 41, 53]. Pioneer trees dispersed by frugivores are often small-seeded and attract numerous dispersers, increasing their potential to be broadcasted widely and thereby colonize farmland and successional habitats near forest [24, 60, 61]. In the absence of isolated trees, such as in the study in Puerto Rico [59], dispersal limitation is severe, but where isolated trees are present, recruitment potential is greatly increased [11, 37]. In the East Usambaras, the legacy trees we sampled were scattered in tea plantations that were enveloped by or bordering forest in this dissected and fragmented landscape, where volant frugivorous animals are frequently seen stopping over either to feed and/or rest intermittently as they move from one forest to another (NJC, HJN, unpublished data). As shown in Los Tuxtlas, Mexico [53], isolated trees in abandoned pastures in a similarly dissected landscape to the East Usambaras, attracted frugivorous bats and birds which were responsible for almost 89% of the seed rain. Similarly, in mixed agriculture abutting Kakamega forest in western Kenya, planted exotic fruit trees *Psidium guavaja* were shown to attract numerous frugivorous birds, including forest dwellers, which may have explained the dominance (82%) of animal-dispersed recruits under these trees [36]. In fact, in isolated, native *Ficus thoningii* trees in mixed agriculture adjacent to Kakamega forest, up to 70 frugivorous bird species visited, 30 of which were forest dependent [61]. While we present strong evidence of the influence of birds and bats in seedling recruitment under legacy trees in the East Usambara tea plantations, seed trap studies separating collections by day and night [e.g.19, 53] would help quantify the contributions that each dispersal vector makes in this process, especially over the long term.

A common recruitment pattern under isolated trees in agricultural and early successional fields adjacent to forest is that pioneers dominate over non-pioneers in the early phases of

succession. As earlier stated, our results add to this trend, and notably, three native pioneer species dispersed primarily by birds, *Bridelia micrantha* (837 recruits), *Macaranga capensis* (538 recruits) and *Shirakiopsis elliptica* (123 recruits) accounted for 54.7% of all pioneer recruits, and were found under almost all legacy trees. A fourth species, *Maesopsis eminii*, an exotic pioneer that is presently a dominant canopy species in the forest and overall landscape, was found under almost all legacy trees with approximately three times more recruits on average under native than exotic legacy trees (Table 3). Specifically, 67.5% of *Maesopsis eminii* recruits were found under bat-dispersed *Milicia excelsa* legacy trees. *Maesopsis eminii* is dispersed by primates, birds, and bats in this landscape [50, 62]. Of these three dispersers, primates are less likely to traverse into the tea plantations (see below), making large birds and bats capable of carrying or swallowing the ~2 cm long diaspores, the primary long-distance dispersal vectors. It is likely these same volant animals are responsible for the transportation of the disproportionately fewer non-pioneers to these legacy trees.

Fleshy fruits with large seeds tend to be eaten by big animals [63]. When these are frugivores, such as the bigger Pteropodid fruit bats and hornbills of the East Usambaras, their larger gape-sizes and/or unique fruit handling abilities facilitates seed dispersal of large diaspores (e.g. for birds: [64]; birds and bats: [50, 63]). Non-pioneers are late successionals, generally characterized by large seeds [e.g. 65], and those that are animal-dispersed are limited by which frugivores are available to disperse their seeds. Of the 202 seedlings of 22 species of non-pioneers, almost two times more recruits were found under native than exotic legacy trees. Some notable non-pioneer recruit species with seed sizes >2–3 cm long [50], include *Cephalosphaera usambarensis*, *Maranthes goetzeniana*, *Parinari excelsa* and *Strombosia scheffleri*. Dispersal limitation of large-seeded tree species into open fields and agricultural matrix is not uncommon [16, 42, 66], and in the dissected and fragmented East Usambara landscape, may be explained by the few medium-sized dispersers like hornbills and fruit-bats that forage or rest in legacy trees. Only two frugivorous primates occur consistently in the submontane plateau, and both of these, the Blue monkey (*Cercopithecus mitis*) and Thick-tailed galago (*Otolemur crassicaudatus*), have rarely been observed in legacy trees or even within the tea plantations (NJC, HJN, unpublished data). In the East Usambaras, elephants are long extirpated whereas bushpigs and baboons occur at very low densities, thus limiting the numbers and types of potential seed dispersers of large-seeded non-pioneer tree species. This is very much unlike in Kibale National Park, Uganda, where Jacob et al. [11] provide evidence that the fruit-eating megafauna like elephants and chimpanzees move from forest into the agricultural matrix and deposit large seeded non-pioneers under planted exotic avocado and mango legacy trees. Therefore, even if hornbills [62] or fruit bats can move large seeds long distances in the study area, such as up to 320.5 m away from the nearest source tree in farmland for the bat-dispersed *Cephalosphaera usambarensis* [50], the low number of non-pioneer recruits is suggestive that these are uncommon events.

## Recruitment composition and overall implications for restoration

Seedling recruitment composition differed among legacy trees in relation to their girth size, fruit type, distance to the nearest forest or their identity as a native versus exotic legacy tree species. We found some overlapping recruit composition between exotic *Grevillea robusta* and native *Milicia excelsa* legacy trees. When animal-dispersed recruits were divided by primary dispersal mode, the results were instructive. The overlapping compositions of bird-dispersed recruits suggests that birds move out of forest and throughout the landscape, using both exotic and native legacy trees almost equally. This is apparent in the three native pioneer species, *Bridelia micrantha*, *Macaranga capensis* and *Shirakiopsis elliptica*, which recruited under

nearly all legacy trees (Table 3). However, for recruit species dispersed by birds-primates-bats, the four primary groupings indicate that at least some avian dispersers use exotic and native legacy trees almost equally, and that fruit bats are likely most responsible for increasing heterogeneity in recruitment composition (Fig 11) [see for example 67]. This is further supported by patterns of recruitment under the 14 exotic legacy trees, where five trees (four *Grevillea robusta* and one *Albizia* sp.) did not have recruits dispersed by birds-bats-primates. On the other hand, *Milicia excelsa*, a bat-dispersed species, showed a distinctively different recruit composition, providing some indirect evidence that fruit bats play a major role here. Evidence for bat seed dispersal and use of natives more so than exotic legacy trees can also be surmised from the distribution of *Maesopsis eminii* recruits. This exotic pioneer is now a dominant canopy species in the forest and overall landscape, and was found under almost all legacy trees in numbers of <40 recruits per tree. In contrast, 67.5% of its recruits were found under bat-dispersed *Milicia excelsa* trees (i.e. a mean ± SE of 95.3 ± 45.7 recruits under *Milicia excelsa* as compared to 12.0 ± 2.2 under the remaining 23 legacy trees). Hornbills, due to their effectiveness as dispersal agents of this species [62], are likely responsible for seed dispersal across all legacy trees. However, more *Maesopsis eminii* recruits under bat-dispersed *Milicia excelsa* trees is likely facilitated when bats come to feed on these legacy trees, or, to process fruits in their dense foliage. Despite uneven sample sizes of the legacy species in this study, the emerging pattern is that native legacy trees seem to recruit different compositions of recruits than exotics. While species richness and recruit abundance was highest under *Grevillea robusta*, the two *Ficus* species, and *Milicia excelsa*, with this trend more apparent in large *Grevillea robusta* trees, it is noteworthy that *Milicia excelsa* and *Ficus* species more specifically, accounted for almost two times more non-pioneer species and recruits (Table 3). If, as shown in this and other studies, fruiting native legacy trees like *Ficus* [53, 54] and *Milicia excelsa* attract frugivores, that contribute new recruit species, including non-pioneers, this can then enhance the overall diversity in forest restoration programs in this region.

An important question in tropical forest restoration, and of much debate, is whether exotic trees should be planted as recruitment foci. Sometimes they can enhance recruitment. For example, in abandoned farmland abutting secondary forest near Kibale National Park, Uganda, Jacob et al. [11] found that fruiting exotic avocado and mango legacy trees not only maintained a unique suite of recruit species compared to areas without legacy trees, but that many recruits were shade-tolerant and dispersed by animals. Furthermore, in the tea plantations bordering forests of the Western Ghats, India, researchers found that tea plantations with *Grevillea robusta* legacy trees had three times more species and 3–30 times the abundance of seeds compared to plantations without this legacy species [35]. About 30% of these recruits were dispersed by birds and mammals [35]. In a subsequent study, tea plantations with *Grevillea robusta* accounted for 92% of the seeds arriving by animal dispersal [41]. While we show that large *Grevillea robusta* had high richness and abundance of recruits, given the high invasive potential of exotic species in the East Usambaras [68, 69], utilizing such species is risky. However, if restoration is permitted in parts of the tea plantation, then preserving legacy trees that are already there, like *Grevillea robusta*, is necessary because of its high recruitment potential.

For restoration purposes, fast growing pioneer species should be planted to favour the rapid development of a canopy. In the East Usambaras, pioneers *Bridelia micrantha*, *Shirakiopsis elliptica* and *Macaranga capensis* recruit in high numbers. Studies on regeneration in Kibale, Uganda, of abandoned farms show that the first two species also showed high recruitment potential (29% *B. micrantha* and 11% of recruits *S. elliptica*). When both these species were planted in a long-term restoration project in the same study area, they made up 95% of stem density at DBHs of ≥ 10 cm after 10 years and after 18 years, *B. micrantha* remained the most

abundant (62% of all planted trees: [70]). While we have an idea of which native pioneers will be successful at developing the initial recruitment template, increasing the presence of late successional species should be concentrated on planting large-seeded non-pioneer tree species which have high dispersal limitation. Furthermore, some tree species may be added by direct seeding [see 10, 71]. When selecting tree species for plantings, as we found in this study, dispersal mode of trees should be taken into account to maximize visitation by the dispersal agents available in the landscape [29]. Among the choices, as we established earlier, would be bat-dispersed *Milicia excelsa* and *Ficus* species, the latter of which have multiple disperser types.

## Supporting information

**S1 Table. Minimum distance of source tree species for seedling recruits under 29 legacy trees in tea plantations in the East Usambara Mts, Tanzania.** Distances are shown to nearest forest with the source tree. Distances in parentheses are of source tree species located closest to the sampled recruit seedlings.
(DOCX)

**S2 Table. Number of species and abundance of recruit species under 29 legacy trees in the East Usambara Mts, Tanzania.**
(DOCX)

## Acknowledgments

Numerous individuals have assisted us on this project, including S. Baruti, R. Bopiah, T. Challange, C. Chulos, M. Cordeiro, T. Hanrahan, A. Holloway, J. Isaay, H. Karata, M. Kijazi, P. Kimali, B. Matunda, B. Mtui, S. Peteler, H. Sengerere, and K. Wentz-Hunter. Logistical support was received from the University of Dar es Salaam (Botany Department), Universidad Autónoma del Estado de Morelos, Amani Parish, Tanzania Serengeti Adventure, Roosevelt University and The Field Museum.

## Author Contributions

**Conceptualization:** Henry J. Ndangalasi, Norbert J. Cordeiro.

**Data curation:** Henry J. Ndangalasi, Tesakiah C. A. Harjo, Clayton A. Pedigo, Rebecca J. Wilson, Norbert J. Cordeiro.

**Formal analysis:** Cristina Martínez-Garza, Norbert J. Cordeiro.

**Investigation:** Norbert J. Cordeiro.

**Methodology:** Henry J. Ndangalasi, Norbert J. Cordeiro.

**Resources:** Henry J. Ndangalasi, Norbert J. Cordeiro.

**Supervision:** Henry J. Ndangalasi.

**Validation:** Henry J. Ndangalasi, Cristina Martínez-Garza, Norbert J. Cordeiro.

**Visualization:** Norbert J. Cordeiro.

**Writing – original draft:** Cristina Martínez-Garza, Norbert J. Cordeiro.

**Writing – review & editing:** Henry J. Ndangalasi, Cristina Martínez-Garza, Norbert J. Cordeiro.

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
