## [Decision Letter · Decision Letter 0]

16 Feb 2021

PONE-D-20-38976

Seedling recruitment under isolated trees in a tea plantation provides a template for forest restoration in eastern Africa

PLOS ONE

Dear Dr. Cordeiro,

Thank you for submitting your manuscript to PLOS ONE. After careful consideration, we feel that it has merit but does not fully meet PLOS ONE’s publication criteria as it currently stands. Therefore, we invite you to submit a revised version of the manuscript that addresses the points raised during the review process.

We look forward to receiving your revised manuscript.

Kind regards,

Christian Vincenot, Ph.D.

Academic Editor

PLOS ONE

Additional Editor Comments:

Dear authors, congratulations on this interesting study. Both reviewers and I would hope to see your work published soon. However, I agree with reviewer 2 on the need to improve the overall quality of the manuscript, which would require many clarifications. I look forward to receiving your revised paper.

Journal Requirements:

"X"

"X"

6.  We note that Figure 1 includes an image of a participant in the study. 

Reviewers' comments:

Reviewer's Responses to Questions

**Comments to the Author**

1. Is the manuscript technically sound, and do the data support the conclusions?

Reviewer #1: Yes

Reviewer #2: Yes

2. Has the statistical analysis been performed appropriately and rigorously? 

Reviewer #1: Yes

Reviewer #2: Yes

3. Have the authors made all data underlying the findings in their manuscript fully available?

Reviewer #1: Yes

Reviewer #2: Yes

4. Is the manuscript presented in an intelligible fashion and written in standard English?

Reviewer #1: Yes

Reviewer #2: Yes

5. Review Comments to the Author

Reviewer #1: A nice study showing the role of isolated trees as the dispersal foci by various animals. Would be indeed very interesting to assess the seed rain during the day (birds) and at night (bats) and the recruitment success of the seeds. It is also possible that many more seed species are brought to the sites but fail to germinate (e.g. due to unsuitable conditions)

Reviewer #2: The manuscript is interesting and has the potential to substantially inform restoration effort in the study region. I do think the impact of the research can be enhanced with some additional revisions, primarily to improve clarity and readability of the manuscript. I have indicated any typos I found and indicated many areas that would benefit from some sort of revision. None of the changes are major, but the number of areas that would benefit from revision is substantial.

6. PLOS authors have the option to publish the peer review history of their article (what does this mean?). If published, this will include your full peer review and any attached files.

Reviewer #1: No

Reviewer #2: No

---

## [Author Response · Author response to Decision Letter 0]

9 Apr 2021

Please find all our responses and clarifications denoted below by asterisks

Additional Editor Comments:

Dear authors, congratulations on this interesting study. Both reviewers and I would hope to see your work published soon. However, I agree with reviewer 2 on the need to improve the overall quality of the manuscript, which would require many clarifications. I look forward to receiving your revised paper.

***Response: We have edited the manuscript following the very constructive and careful criticisms of reviewer 2. We consider all the changes made add more clarity to this paper. If at all possible, would you be so kind as to privately inform this reviewer that the quality of the critique and suggestions made were outstanding and greatly improved this Mss. It was also gratifying to receive a positive comment about a large section of the Discussion.

Reviewer #1: A nice study showing the role of isolated trees as the dispersal foci by various animals. Would be indeed very interesting to assess the seed rain during the day (birds) and at night (bats) and the recruitment success of the seeds. It is also possible that many more seed species are brought to the sites but fail to germinate (e.g. due to unsuitable conditions)

***Response: We sincerely thank Reviewer 1 for their kind, positive remarks. It is absolutely possible that more seed species could arrive under legacy trees and fail to germinate. However, the point of this paper is to develop a recruitment template for future restoration in the area, so know what seedling species have established is critical towards this goal. We hope to have a MSc or PhD student follow through with seed traps in the future.

Reviewer #2: The manuscript is interesting and has the potential to substantially inform restoration effort in the study region. I do think the impact of the research can be enhanced with some additional revisions, primarily to improve clarity and readability of the manuscript. I have indicated any typos I found and indicated many areas that would benefit from some sort of revision. None of the changes are major, but the number of areas that would benefit from revision is substantial.

***Response: We are most grateful for all the heavy criticism, but also providing suggestions on how to handle some of the revisions. We revised every single recommendation and feel that the manuscript has improved markedly. 

PONE-D-20-38976

Seedling recruitment under isolated trees in a tea plantation provides a template for forest restoration in eastern Africa

The manuscript is a very interesting investigation into passive tree recruitment under legacy trees in a tea plantation. An understanding of how legacy tree type (native or exotic) influences recruitment and the roles of dispersal mechanism and successional status on recruitment has the potential to inform conservation efforts and focus restoration activities in the area. The data analyses are appropriate in my opinion and I did not notice any flaws in logic. After some revisions, I believe the manuscript would be considered relevant, informative, and useful to many readers of Forest Ecology and Management. 

***Response: Many thanks. 

Comments:

The manuscript would benefit substantially if revised extensively for clarity. The writing style is cumbersome at times and creates needless complexity given the topic and analyses. The cumbersome style is not present in all portions of the manuscript, but in general it makes the manuscript challenging to read. I will not indicate every sentence that I consider likely to benefit from rewording for clarity, but I will try to point out some areas that the authors should consider revising.

***Response: Unless a clarification was needed, we have handled all the changes suggested below.

Line 22 – less extensive might be a better word choice than cheaper

Line 25 – comprised of rather than represented by

Line 28 – 98% of what recruits? Pioneer or non-pioneer or all recruits?

Line 31 – bat-dispersed rather than seed dispersal by bats

Line 32 – changes in comparison to what

Line 34 – toward seems to imply direction, in regards to might be more appropriate

Line 35 – vast usually implies space, a different word might be more appropriate

Line 36 – the sentence that begins with “in terms of isolated….” Needs to be reworded for clarity

Line 40 – what type of animal diversity are you referring to?

Line 47 – extra “the” in the sentence that should be deleted

Line 53 – The sentence that begins “Forest fragmentation in this range…..” need to be reworded

Line 57 – inevitable implies that conservation efforts are a waste of time

Line 63 – The sentence beginning with “International” needs to be reworded for clarity

Line 71 – evaluated seems more appropriate than measured

Line 72 – This last statement might need to be qualified with an acknowledgement that passive recruitment and the template is dependent on landscape characteristics

***Response: A good point.

Line 74 – limitation or limitations?

Line 89 – animals and delete “species or taxa”

Line 91 – delete even

Line 95 – replace while with and

Line 100 – If isolated trees improve environmental conditions and increase tree recruitment why would you expect more pioneers than non-pioneers? I understand the logic, but the writing sounds as if you are arguing opposite points. Reword or just reorganize paragraph for clarity. 

***Response: We clarified this further.

Line 102 – retained rather than “left”

Line 105 – delete may

Line 109 – delete “also” and “the middle of” 

Line 112 – change “might be useful to” to may

Line 117 – We follow Jacobs et al. and Chetana et al. in our use of the term legacy trees to refer to isolated trees which are remnants from the past.

Line 121 – limitations

Line 121 – indicate rather than show

Line 124 – delete “exotic or native”

Lines 123-126 – reword for clarity

Line 129 – based on tree size….

Line 132-134 – reword for clarity and/or readability

Line 142 – above sea level rather than abbreviating

Line 142 – why is mm the unit used rather than alternative cm

***Response: We prefer the unit mm because that is the international standard.

Line 143 - you mention seasonal rainfall, then persistent rainfall most of the year

Line 146 – no need for the decimal if you already indicate approximate

Line 148 – tea plantations don’t cause the destruction and fragmentation, they are the impetus for those changes but not the mechanism for the changes

Line 150 – reorganize sentence. Protected as the 8360 ha Amani Nature Reserve

Line 168 – plantations

Line 168 – What is “(e.g. 34)” I don’t know what this 

Line 187 – how were tree heights measured

Line 189 – delete closest

Line 190 – can this sentence be worked in with the previous sentence. As is, it seems a little out of place and an afterthought

Line 196 – reword. “where the leaves were freshly..” with leaves freshly emerged from 

Lines 193 – 199 – paragraph does not flow and should be reorganized

Line 251 – I think it is beneficial to include a brief justification for and/or establishment as appropriate for analytical methods with citations when possible. It is also appropriate to provide readers with a very general guidance on statistical outputs from methods and how they should be interpreted very broadly

***Response: A good point. We have made the recommended changes.

Line 252 – a number would be more beneficial than “few”

Line 254 – change “not included in” to excluded from

Line 257 – delete also

Line 262 – delete also

Line 294 – Fourteen recruits (0.5%) were … 

Line 297 – (2738 seedlings; X%), whereas the remainder were abiotically…..

Line 301 – clarity was compromised throughout the manuscript when semicolons were used. I think revision of all these sentences and splitting into multiple sentences is typically warranted. Brevity should not trump clarity and readability in my opinion. 

***Response: We have removed numerous semi-colons and split these sentences. We agree that readability has improved markedly.

Line 305 – delete all

Lines 312-315 – reword and possibly split into multiple sentences. Delete “and of great interest”

Line 326 – change “pin-point” to something more appropriate. Possibly to identify preliminary trends

Line 326 – delete “for” 

Line 326 – change grouped to categorized, delete these, reword

Lines 329-349 – align by decimal point or similar method

***Response: We made all format changes to tables.

Line 351 – Delete “for the PCA” and reword

Line 367 – possibly change longer distance away to farther

Line 369 – change where possibly to in which or similar

Line 442- comma after agroforestry and whenever you have a list of 3 

Lines 443- 447 – reword this long sentence for clarity

Line 448 – delete exotic and native

Line 457 – reword first sentence. Likely split into 2 sentence and clarify what you are trying to say in last part of the sentence.

Line 468 – factored? Clarify what you mean by factored in this sentence

Line 472 – comma after agricultural

Line 481 – clarify or reword the 56-99% dispersal limitation phrase

Line 482 – 487 – long cumbersome sentence that should be reworded and split into multiple sentences for clarity

Line 489 – what does the phrase “apparently planted source trees”

Lines 494-559 – really well written section of the manuscript

***Response: Thank you so much! 

Line 513 – consider changing high 82% to “explained the dominance (82%) of animal-dispersed

Line 524 – replace formed with accounted for 54.7%

Line 525 – replace across with under or associated with or similar

Line 527 – replace just about with approximately three times more recruits…

Line 528 – replace its with M. eminii

Line 536 – reword this first sentence. Replace like with such as and reword for clarity

Line 563 – replace given with based on in relation to

Line 563 – does size relate to seed size or tree height or dbh? 

Line 568 – the phrase “were very telling” seems to conversational so consider rewording

Line 570 – delete an example of where. This is apparent in the three

Line 573 – Is it a given that birds use exotic and native trees equally OR do some birds use them equally (those the disperse certain tree seeds) and others exhibit preferences (those that disperse fruits from the birds+bats+primates category). I don’t know enough about the trees being considered, the categories you placed trees into, or the areas you work. However, it seems like a potential overreach based on just what you report to assume all birds are using exotic and native legacy trees almost equally. You provide some evidence and reasoning in the paragraph below but I am not sure you are not making inferences beyond your data or the data presented.

***Response: An excellent point. We do not think it was intentional to state that all birds use exotic and native trees equally. We altered the wording to avoid this overstatement.

Line 580-582 – This sentence is a little unclear regarding the numbers mentioned and also what you want the reader to take from the sentence. Maybe indicate the average number under M. excelsa rather than or in addition to the percentage to allow for a better comparison.

***Response: We added the mean number of Maesopsis eminii recruits as suggested.

Line 592 – delete will 

Line 593 – delete will

Line 593-594 – reword “have the capacity to increase the potential to enhance”

Line 600 – missing something here. Maybe “not” is a typo and should be “to”

Line 603 – missing something legacy trees three times more??

Lines 606-607 – reword this sentence, the last part after the comma is awkward 

Line 612 – is there a more appropriate term than necessarily encourage – possibly without risk or risk of ecological consequences

Line 615-617 – this sentence almost argues that there is little to no value of planting pioneer species as a restoration method. If the goal is restoration in these plantation, it might be better to phrase this in terms of restoration activities that might be favored over planting. For restoration purposes, reducing competition or eliminating the practice of annual seedling removal under legacy trees may be sufficient for pioneer species given that pioneers arrive and recruit passively. 

Line 625 – insert species after successional

FIGURES – abundance is misspelled in several locations within the figures. Either on the axes or elsewhere. 

***Response: We made all the changes.

---

## [Editor Report · Decision Letter 1]

15 Apr 2021

Seedling recruitment under isolated trees in a tea plantation provides a template for forest restoration in eastern Africa

PONE-D-20-38976R1

Dear Dr. Cordeiro,

We’re pleased to inform you that your manuscript has been judged scientifically suitable for publication and will be formally accepted for publication once it meets all outstanding technical requirements.

Kind regards,

Christian Vincenot, Ph.D.

Academic Editor

PLOS ONE

Additional Editor Comments (optional):

Please make sure to thoroughly check the manuscript for typos or other minor mistakes (also in the references) and, if present, address them when provided with the proof. Please note that PLOS does NOT perform quality check (incl. typo/refs check) on manuscripts, so this is entirely the responsibility of authors.
---

## [Editor Report · Acceptance letter]

23 Apr 2021

PONE-D-20-38976R1 

Seedling recruitment under isolated trees in a tea plantation provides a template for forest restoration in eastern Africa 

Dear Dr. Cordeiro:

I'm pleased to inform you that your manuscript has been deemed suitable for publication in PLOS ONE. Congratulations! Your manuscript is now with our production department. 

Kind regards, 

on behalf of

Dr. Christian Vincenot 

Academic Editor

PLOS ONE